# Effect of Polyphenol Supplementation on Memory Functioning in Overweight and Obese Adults: A Systematic Review and Meta-Analysis

**DOI:** 10.3390/nu16040474

**Published:** 2024-02-06

**Authors:** Sara Farag, Catherine Tsang, Emad A. S. Al-Dujaili, Philip N. Murphy

**Affiliations:** 1Department of Psychology, Edge Hill University, Ormskirk L39 4QP, UK; 24058416@edgehill.ac.uk (S.F.); murphyp@edgehill.ac.uk (P.N.M.); 2Department of Applied Sciences, Northumbria University, Newcastle upon Tyne NE1 8ST, UK; catherine.tsang@northumbria.ac.uk; 3Centre for Cardiovascular Science, Faculty of Medicine and Veterinary Medicine, Queen’s Medical Research Institute, University of Edinburgh, Edinburgh EH16 4TJ, UK

**Keywords:** cognition, memory, obesity, polyphenols, meta-analysis, systematic review

## Abstract

Negative health consequences of obesity include impaired neuronal functioning and cell death, thus bringing the risk of impaired cognitive functioning. Antioxidant properties of polyphenols offer a possible intervention for overweight people, but evidence for their effectiveness in supporting cognitive functioning is mixed. This review examined evidence from randomized controlled trials concerning the effect of polyphenols on tasks requiring either immediate or delayed retrieval of learned information, respectively, thus controlling for differences in cognitive processes and related neural substrates supporting respective task demands. Searches of the PubMed/Medline, PsycInfo, and Scopus databases identified 24 relevant primary studies with N = 2336 participants having a BMI ≥ 25.0 kg/m^2^. The participants’ mean age for the 24 studies exceeded 60 years. Respective meta-analyses produced a significant summary effect for immediate retrieval but not for delayed retrieval. The present findings support a potential positive effect of chronic supplementation with polyphenols, most notably flavonoids, on immediate retrieval in participants aged over 60 years with obesity being a risk factor for cognitive impairment. We recommend further investigation of this potential positive effect in participants with such risk factors. Future research on all populations should report the phenolic content of the supplementation administered and be specific regarding the cognitive processes tested.

## 1. Introduction

Severe cognitive impairment, or dementia, is a growing public health concern affecting millions of people globally [1]. The disorder is characterized by a progressive decline in several cognitive domains that interferes with daily functioning [2]. Mild cognitive impairment (MCI) is seen as a transitional phase between a normal brain state and dementia [3]. Recent evidence has linked obesity to the development of cognitive impairment [4,5], with a higher body mass index (BMI) in middle-aged people being associated with the risk of developing Alzheimer’s disease and vascular dementia being elevated [6,7]. The mechanism responsible for this may be the association of obesity (especially abdominal fat) with chronic inflammation and oxidative stress.

Increased adiposity has been linked to chronic inflammation through the excessive secretion of pro-inflammatory adipokines, namely tumor necrosis factor alpha (TNFα) and interleukin-6(IL-6) [8,9]. Additionally, chronically elevated free fatty acids have been found to cross the blood–brain barrier and induce inflammation in the brain, with the consequence of these processes being the generation of oxidative stress and neuroinflammation [10,11]. This is of great importance, as obese populations have also been found to have imbalanced oxidative/antioxidant defenses (e.g., enzymes), which facilitate these harmful effects further [10,12], thus reinforcing the development of neuronal damage, gliosis (fibrosis of brain tissue), neuroinflammation, and neuronal cell death [13]. In turn, the consequent likelihood of cognitive impairment is increased [14,15].

The role that inflammation and oxidative stress plays in neural damage, especially in obese and overweight populations, imposes the need to seek interventions that could control such processes to slow and prevent cognitive decline. One possibility is the use of dietary constituents such as polyphenol [14]. These are secondary plant metabolites widely dispersed in the human diet, with fruits, vegetables, tea, and red wine providing abundant sources [16], which have been shown to exert antioxidant and anti-inflammatory effects [14,17]. Much evidence has been reviewed regarding the effectiveness of polyphenol-rich foods for preventing or slowing the progression of cognitive impairment caused by cerebrovascular and neurodegenerative disorders [18], and several randomized controlled trials (RCTs) have also shown their association with enhanced cognition in overweight participants [19,20,21]. However, other studies have failed to show similar positive effects [22,23,24]. With the existing literature having several limitations (e.g., small sample sizes and variation in polyphenol dose and composition, type, and study design), it is currently difficult to reach a comprehensive conclusion regarding the effect of polyphenols on cognition. Furthermore, only a limited number of studies have included overweight and obese participants. Consequently, the present review and meta-analyses focused only on studies including overweight and obese populations. 

In keeping with earlier reviews of polyphenol supplementation [25,26], care was taken to identify cognitive tasks that drew upon specific psychological processes to make precise conclusions regarding the effects reported. This review examined the effects of polyphenol supplementation on immediate and delayed memory retrieval performance, respectively. The short-term storage of material, for example, in immediate recall or recognition tasks, is related in the psychological literature to the construct of working memory (WM), which comprises both short-term storage and additional task-related processing, with the latter being part of the executive functioning that enables adaptation to and management of the immediate environment [27,28]. Although the longer-term storage of material is not part of WM itself, the two-way passage of material from WM to and from long-term memory is assumed theoretically [29] and has been demonstrated empirically [30,31,32]. By examining the results of polyphenol supplementation for immediate and delayed retrieval tasks separately, the intention was to control processing differences between the tasks and the neural substrates related to these [33,34,35]. Task differentiation such as this is consistent with good practice when conducting meta-analyses regarding generating summary effects that are meaningful [36,37]. 

The effectiveness of polyphenol supplementation for the support of memory functioning in overweight and obese participants was the focus of the present systematic review, with meta-analyses, of published RCTs. Its principal objective was to elaborate existing knowledge concerning whether polyphenols are effective in improving performance on tasks requiring either the immediate or delayed retrieval of learned information, respectively.

## 2. Materials and Methods

A research protocol (see Appendix A) was used for this examination of the effect of polyphenols on memory functions. In addition, the Preferred Reporting Items for Systematic Reviews and Meta-analysis (PRISMA) guidelines were followed in the present study [38].

### 2.1. Data Sources and Search Strategy

A comprehensive systematic search of randomized controlled trials (RCTs) published in the English language was conducted electronically in PubMed/Medline, PsycInfo, and Scopus databases up to August 2023. The search terms contained both a component for polyphenols and a component for memory functioning. The polyphenol component comprised one of the following 15 keywords or terms: polyphenol, pomegranate, flavonoids, polyphenolic compound, polyphenolic compounds, isoflavone, flavanol, phytoestrogen, resveratrol, ellagitannin, ellagic acid, punicalagin, or anthocyanins, proanthocyanidin, and proanthocyanidins. These 15 keywords were initially searched independently, with the consequent results then being combined using the operator instruction “OR”. These results were then combined, using the operator instruction “AND”, with the results from the searches using the keywords and terms comprising the memory functioning component. This latter component comprised one of the following 36 keywords or terms: mild cognitive impairment, MCI, cognition, cognitive performance, cognitive function, brain function, memory, neuroimaging, neural, magnetic resonance imaging, MRI, fMRI, grey matter, gray matter, brain structure, electrophysiology, EEG, event-related potential, neuroblast, cerebral blood flow, CBF, regional perfusion, pulsatility index, transcranial doppler, TCD, near-infrared spectroscopy, NIRS, total hemoglobin, oxygenated hemoglobin, oxy-Hb, deoxygenated hemoglobin and Deoxy-Hb, immediate recall, delayed recall, long term memory, or short term memory. These 36 keywords or terms were initially also searched independently before being combined with the operator instruction “OR”. The RCT filter was then used to complete the searches. 

It should be noted that tests of immediate or delayed recall and recognition, for example, are generally administered as part of a battery of cognitive tests, rather than being mentioned themselves in abstracts. By using search terms such as “cognitive performance”, it was, therefore, possible to identify studies for which further examination would indicate their relevance or lack of it to this review. To identify additional relevant studies, the search was supplemented by manually cross-matching reference lists from the respective databases and searching citations of relevant studies from review articles. Duplicate publications were removed by RefWorks with additional manual checking prior to evaluation against the inclusion and exclusion criteria. 

### 2.2. Study Selection

The PICOS criteria shown in Table 1 were applied as inclusion criteria for the selection of studies. Additional inclusion criteria were that the studies reported results of primary research in peer-reviewed journals, using the English language. Studies were accepted if their participants reported mild cognitive impairments (MCI) or subjective memory problems, but these were not a condition for acceptance.

Studies published in languages other than English were removed in accordance with the exclusion criteria, as were studies of other aspects of cognitive functioning, studies that had recruited participants diagnosed with severe cognitive impairment or dementia, animal studies, in vitro studies, and case studies. Other types of studies excluded were those where no supplementation had been given or that had administered other supplements in addition to polyphenols. Studies with serious methodological deficiencies, such as the absence of a placebo condition or the lack of random participant allocation to conditions, were also excluded. Encyclopedia entries and book chapters were excluded due to the likelihood of variation in peer-review practices, and conference or workshop presentations were excluded unless appropriate peer-review processes were apparent. The format of result presentation was also an important consideration for inclusion, as results on specific memory performance tasks had to be explicitly identifiable within the paper, available online as supplementary material, or available through correspondence with the authors. Consequently, papers that provided memory task results only in the form of composite scores that incorporated results from other tasks were excluded.

### 2.3. Data Extraction

The data extracted from each study were author(s) name, year of publication, study design, treatment characteristics, dosage of polyphenol-rich supplements, characteristics of the treatment and placebo groups, and intervention duration. Outcome data were extracted concerning differences in immediate and delayed retrieval performance, respectively, between polyphenol supplementation conditions and control conditions. Although the distinction between these two forms of retrieval was sometimes explicitly clear, as with immediate and delayed recall procedures, it was sometimes necessary to define retrieval based on whether a participant’s readiness for the retrieval of a stimulus item had been terminated or not. For example, the N-back task was classified as requiring immediate retrieval in this review, as participants had to maintain the availability of items for retrieval up to four stimulus items later. Details of tasks not requiring immediate or delayed retrieval of items administered specifically for retrieval were not extracted.

### 2.4. Quality Assessment

The methodological quality of selected studies was assessed using the Jadad scale [39], which has been widely used in the literature for this purpose in relation to RCTs. Each paper was reviewed in response to questions concerning its reported use of randomization and blinding and the reporting of withdrawal/dropout rates. Uncertainties in these areas were resolved through consensus amongst the authors. A maximum score of 5 (the sum of awarded points) was awarded only where all criteria had been satisfied. Scores of 3 or more have been judged by previous reviews as an indication of “high quality” in previously published reviews, whilst scores less than 3 have been taken to indicate “low quality” [40]. Jadad scores were not used as a criterion for sample inclusion in the present study.

### 2.5. Meta-Analytic Strategy

Comprehensive Meta-Analysis software (CMA for Windows, Version 3, Biostat, Englewood, NJ, USA, 2013) was used for all the meta-analyses reported here. With all included studies being RCTs, and in accordance with guidance received from external advisors, post-intervention scores were entered as data for both intervention and placebo conditions, respectively. It was necessary to protect the integrity of summary effect sizes where studies reported more than one relevant dependent variable, e.g., [41], where different doses of a supplement were compared to either a baseline or placebo condition, e.g., [42], and/or the study reported more than one post-administration duration for comparisons between conditions, e.g., [43]. In these situations, the mean effect size of the multiple comparisons was taken. With only one value for each primary study being present in the meta-analyses, risks of distortion from between-participants studies with parallel arms were avoided [44].

The meta-analytic strategy comprised three levels of analysis, with level one analyses comprising all primary studies in the sample. For level two, separate analyses were conducted for between-participants RCTs (BTW-P) and crossover trials. This was done as a precaution concerning potential distortions due to the presence of both forms of trial in the level one analyses [44]. Following the procedure described by Borenstein et al. [36], sensitivity analyses were conducted for the level two meta-analyses by removing one study at a time. If either the level one or level two meta-analyses yielded significant summary effects, level three meta-analyses were conducted, where possible, on studies having administered the same polyphenol. These meta-analyses also maintained the distinction between BTW-P RCTs and crossover trials. However, studies were excluded from level three meta-analyses if they had failed to report the polyphenol content of the supplementation administered.

Consistent with good practice for meta-analyses [36,37,38,39,40,44,45], the level one and level two meta-analyses used random-effects models as an a priori choice rather than basing the choice of model on results of tests concerning the heterogeneity of effect sizes. Borenstein et al. [45] criticize the use of heterogeneity tests in this way due to their low statistical power in calculating the *Q* statistic, which may lead to a lack of sensitivity to a distribution of true population effect sizes from which the samples of data gathered by the primary studies will have originated. The primary studies in the level one and level two meta-analyses reported here had administered a range of polyphenols, so that a consequent a priori foundation existed to assume that a distribution of true effect sizes had been sampled, thus necessitating use of a random-effects model. As polyphenol type did not differ between studies in the level three meta-analyses, fixed-effect models were initially used, with this assumption then checked against the *Q* statistic generated.

To minimize any distortions from small sample sizes in some of the primary studies, Hedges’ *g* was employed as the effect size metric in all meta-analyses. Where the primary studies reported better performance for supplementation conditions than for placebo conditions, outcomes were coded as positive. Outcomes were coded as negative if the opposite was the case. An alpha level of *p* ≤ 0.05 was employed in all analyses. The following tests for publication bias were employed for the level one meta-analyses: Rosenthal’s fail-safe *N* statistics, Begg’s and Mazumdar’s rank correlation test (Kendall’s *S* statistic *P*-*Q*) [46], Egger’s linear regression test [47], and Duval’s and Tweedie’s trim and fill test [48]. However, Rosenthal’s fail-safe *N* was only relevant and therefore used if the level one result was significant. For nonsignificant level one results, it remained relevant to establish if the obtained summary effect size had been biased by the presence of primary studies with smaller participant samples reporting larger effect sizes, compared to effect sizes reported by studies having larger participant samples. Forest plots were used where square boxes represent the obtained effect sizes for individual primary studies, with their attached lines representing 95% confidence intervals. Diamond shapes represent the summary effect size for the meta-analysis.

## 3. Results

### 3.1. Study Selection

A total of 3330 possible publications were identified using the search strategy reported. Removal of duplicates left 1328 publications, and the further removal of nonclinical trials (e.g., animal and cell culture studies), literature reviews, book chapters, and clinical studies continuing to recruit at the time of the search left a sample of 309 articles. A first screening by titles and abstracts for eligibility against inclusion and exclusion criteria led to a provisional list of 129 published research articles. After reviewing the full texts of these articles, 99 were excluded, leaving 30 studies to be considered for inclusion (see Figure 1). Careful examination of the remaining 30 studies identified 24 that met the inclusion criteria and examined the effect of polyphenols on memory functions. A summary of this process is shown in Figure 1. All studies in the final sample used a conventional alpha level of *p* < 0.05.

### 3.2. Study Design and Participant Characteristics

The data extracted from each primary study in the sample are shown in Table 2.

The 24 studies in Table 2 comprise 14 reported findings from both immediate and delayed retrieval tasks, with five additional studies that reported relevant findings for immediate retrieval [19,20,23,42,51] and five additional studies that reported relevant findings for only delayed retrieval [52,55,56,57,64]. Table 1 shows that five of the 24 studies reported findings concerning the acute effects of a single dosage up to 8 h after administration [19,41,47,49,52], with the remaining 19 studies reporting findings concerning chronic administration for up to 2.5 years in one case. 

The included studies comprised 2336 participants. Only five of them were crossover trials, with sample sizes ranging from 14 to 42 participants, and a mean of ages for participants of 59.70 years (*SD* = 8.96 years). The remaining 19 studies with between-participants designs reported total sample sizes ranging from 20 to 657 participants. Four of these studies [19,20,42,58] reported more than one treatment comparison to the same placebo group, resulting in 24 comparisons in total. The mean age for the placebo groups was 64.99 years (*SD* = 5.40 years) and 64.87 years (*SD* = 6.26 years) for the treatment groups. Paired-samples *t*-tests (i.e., treating each study as one participant) comparing age and sample size, respectively, between the control and intervention groups in these 18 studies showed no significant differences (*t* < 1 in both cases). One study reported median rather than mean ages and was not included in the present age calculations [19]. 

Twenty-two studies in the sample adopted a randomized double-blind design. Of the two exceptions, one study employed an observed blinded parallel group randomized design [24] and the other [41] only reported blinding for the participants in their crossover trial. Polyphenol supplementation in the primary studies took a variety of forms. Six studies, one of them [58], used resveratrol extract with a dose ranging from 75–1000 mg per-day. Five other studies (e.g., [21]) used blueberry, either as a concentrate or a beverage, which allowed for a total daily intake of anthocyanins from 269 to 579 mg per day, whilst Cook et al. [50] administered blackcurrant extract with a daily intake of 210 mg of anthocyanins. Soya-based supplementation was used by two studies in the form of either soya milk and supplement (Fournier et al. [42]: 70 mg of isoflavone per day) or of isoflavone-rich soy protein powder (Henderson et al. [53]: 91 mg of isoflavones per day). Other interventions included polyphenol-rich pomegranate juice (Siddarth et al. [61]: 368 mg of punicalagins per day), green oat extract (Kennedy et al. [43]: either 800 mg or 1600 mg per test session), cosmos caudatus extract (7.41 mg/day total polyphenols) by You et al. [51], spearmint extract (14.5% rosmarinic acid) by Herrlinger et al. [20], curcumin supplements by Rainy-Smith et al. [62] (1500 mg per day), and orange juice with a daily intake of 272 mg flavonoid [49].

Several studies included in the current systematic review and meta-analysis did not mention details regarding their supplement polyphenol content. For instance, commercially prepared curcumin supplement was used by Cox et al. [63] (80 mg per day) without describing the exact phenolic content of this supplementation. Similarly, a walnut diet (30–60 g/day) was utilized [24], and the acute effects of turmeric, cinnamon, or both were examined in one study [19], without the phenolic content being reported. However, the literature described the phenolic content of similar foods/diets, such as curcumin and turmeric [62,65,66], walnut [67], and cinnamon [68]. Nonetheless, it should be noted that the polyphenolic content of these studies [19,24,63] included in this review might differ from those reported in the literature, and for that, it is important to detail the phenolic content of the supplement used. 

### 3.3. Cognitive Tasks and Cognitive Batteries

Different cognitive test batteries were used by the primary studies, seven of which implemented a single test battery, these being the Cambridge Neuropsychological Test Automated Battery (CANTAB: [58]), the cognitive function test battery (CogState Ltd., Melbourne, Australia: [21,62]), a comprehensive neuropsychological battery on the Neurology^®^ web site [53], the Computerized Mental Performance Assessment System (COMPASS: Northumbria University [43,63]), and the National Institutes of Health Toolbox (NIH-Tool Box: [59]). The remaining studies used a battery of validated cognitive tests to assess the effect of polyphenol-rich supplementation on a variety of memory retrieval procedures, as shown in Table 2.

### 3.4. Methodological Quality of Studies 

The results of the methodological quality review using the Jadad scale are shown in Table 3. A score of less than 3 was obtained for only one study in the sample. The sample mean score was 4.08 (*SD* = 0.82). Eight studies received the maximum score of 5, indicating that all criteria had been met, and eleven studies obtained a score of 4, due to an absence of details concerning the procedure for either randomization or blinding or an absence of details concerning participants’ outcomes. Four of the remaining studies were judged to have met only three of the scoring criteria. Overall, however, based on the mean score, the quality of the primary studies was judged as being good to excellent. 

### 3.5. Effect of Polyphenol-Rich Supplementation on Memory Retrieval

Only seven of the primary studies in Table 1 reported significant effects in memory retrieval tasks relevant to this review. Four of these studies reported a positive effect on at least one measure of immediate retrieval [19,20,41,61], two studies reported a positive effect on at least one measure of delayed retrieval [43,57], and one study reported an effect on both [60]. All seven studies administered different phenolic compounds. 

Although Dodd et al. [41] reported an improvement in immediate word recognition following blueberry supplementation, two other studies reported no effect for blueberry supplementation on other measures of immediate retrieval [21,23]. Dodd et al. [41] administered the highest phenolic content per supplementation dose of these three studies (579 mg), which was administered as a one-off dose to measure its acute effects. Supplementation in the other two studies lasted for 16 and 12 weeks, respectively. Collectively, these studies reported levels of the related compounds of anthocyanin, anthocyanidin, procyanidin, and cyanidin 3-glucoside in their supplementation regimes. Anthocyanin also comprised the phenolic content of the blackcurrant supplementation administered for 7 days by Cook et al. [50], a lower phenolic dosage than Dodd et al. [41], with no acute effects on immediate memory retrieval being reported. However, anthocyanin was present in the chronic pomegranate juice supplementation administered by Siddarth et al. [61], with better immediate recall reported at both 6- and 12-month follow-ups. This study had a slightly higher phenolic content per dose (i.e., 588 mg per day for 12 months) than that reported by Dodd et al. [41] for a one-off dose. Lee et al. [19] reported an improved effect on immediate retrieval relating to turmeric but not cinnamon administration, with no specific information on the phenolic content of the supplementation. Furthermore, similar positive effects on immediate retrieval were reported by Evans et al. [60] and Herrlinger et al. [20] following the chronic administration of resveratrol (75 mg) and spearmint extract (900 mg) for 14 and 12 weeks, respectively. No other study in Table 1 investigated turmeric, spearmint, or resveratrol on immediate retrieval to permit comparisons with these results.

In addition to Evans et al. [60], two other studies reported an enhancement of delayed retrieval, with this following acute administration of green oat extract (GOE: [43]) and chronic administration of resveratrol [57], respectively. No other study in Table 1 administered GOE to permit comparisons with Kennedy et al. [43]. Resveratrol was also chronically administered in two studies reporting no effects for either immediate or delayed retrieval. Anton et al. [58] reported no significant effects with the respective doses of 300 mg and 1000 mg per day for 90 days, whilst Thaung Zaw et al. [59] used a relatively low dose of 75 mg (which was like the dose used by Evans et al. [60]) for 12 months and reported no significant effect. An additional two studies reported no effects related to resveratrol for delayed retrieval having been respectively administered 520 mg daily for 26 weeks [55] and 550 mg daily for 26 weeks [56]. Consequently, the dosage of Witte et al. [57] of 520 mg daily for 26 weeks, which was related to enhanced delayed retrieval, was not large in comparison to the studies reporting no effects for memory retrieval. Despite the sample-appropriate language difference in the tasks administered by Witte et al. [57], these tasks do not appear to have differed from those used in the other resveratrol studies in any way that would be expected to affect retrieval performance.

Enhanced memory retrieval was not related to chronic isoflavone supplementation through either soya milk or soy protein powder in three studies [42,53,54]. Kreijkamp-Kaspers et al. [54] reported the highest daily phenolic dosage of these studies (99 mg per day for 12 months). Chronic curcumin supplementation was not related to effects concerning memory retrieval in two studies [62,63]. The nonsignificant findings of these five studies concerned both immediate and delayed retrieval, except for Fournier et al. [42], which examined only immediate retrieval. 

### 3.6. Meta-Analyses Results

#### 3.6.1. Immediate Retrieval

Sixteen studies were included in the level one random-effects meta-analysis for immediate retrieval. Three studies in the review sample were excluded from the analysis because of the lack of statistical information necessary for inclusion [19,59,63]. The problems included a lack of clarity as to whether standard deviations or standard errors were being reported, the reporting of results in graphical form only, and the use of only a general statement to report nonsignificant results without any statistical detail. The weighted mean effect size for this random-effects analysis was significant (Hedges’ *g* = 0.170; 95% CI 0.007 (lower) to 0.333 (upper); *z* = 2.044), *p* = 0.041; *Q* (*df* = 15) = 29.828, *p* = 0.013, *I*^2^ = 49.712, tau squared = 0.047). These results are illustrated by the forest plot in Figure 2, which shows better immediate retrieval with polyphenol supplementation than with placebo administration.

For publication bias, Duval’s and Tweedie’s trim and fill procedure did not trim any studies. The results for Kendal’s tau with continuity correction (tau = −0.008 ns.) and for Egger’s regression procedure (*t* (14) = 0.531, ns.) were evaluated as two-tailed and were nonsignificant. Rosenthal’s fail-safe *N* for this meta-analysis indicated that 20 studies would be needed to make its result nonsignificant. Consequently, these results indicated no evidence of significant publication bias in the results of this level one meta-analysis (see the funnel plot in Figure 3).

Both duration of supplementation and daily polyphenol dose in respective studies were regressed against the study effect sizes. Neither of these analyses produced significant results (*p* = 0.946 and 0.695, respectively, and *R*^2^ = 0.00 in both analyses).

##### Level Two Meta-Analyses for Respective RCT Designs

Between-participants design RCT (BTW-P). Between-participants design RCT (BTW-P). The random-effects model yielded significant results (*g* = 0.226, *z* = 2.209, *p* = 0.027) with a significant heterogeneity (*Q* (*df* = 11) = 27.77, *p* = 0.004, *I*^2^ = 60.399, tau squared = 0.064). The main results of this analysis are summarized in Figure 4. Consequently, immediate retrieval was better with polyphenol supplementation than with placebo administration.

Crossover trials (CO). The results of the random-effect model for the four crossover trials were nonsignificant (*g* = −0.007, z = −0.052 ns.; *Q* (*df* = 3) = 0.640, ns., *I*^2^ = 0.000, tau squared = 0.000). This value of tau squared indicates that the random effects and fixed-effect models were equivalent for this sample [45]. The results of this analysis are also summarized in Figure 4.

##### Level Three Meta-Analyses by Polyphenol Type for Immediate Retrieval

The level three meta-analyses grouped the primary studies by polyphenol type where this was possible. These analyses did not include primary studies that had not explicitly reported the type of polyphenol administered. These studies were labeled as unclassifiable in the present review. As previously explained, level three meta-analyses initially used fixed-effect models due to the common use of a particular polyphenol [45].

Flavonoids were administered by seven studies, of which three were BTW-P RCTs, and four had used crossover designs. The results for the BTW-P RCTs were borderline nonsignificant (*g* = 0.491, *z* = 1.921, *p* = 0.055; *Q* (*df* = 2) = 5.693, ns., *I*^2^ = 64.868, tau squared = 0.373). Figure 5 summarizes the results of the meta-analysis. The results for the crossover studies were nonsignificant (*g* = −0.007, *z* = −0.052, ns.; *Q* (*df* = 3) = 0.640, ns., *I*^2^ = 0.000, tau squared = 0.000). Figure 5 also summarizes the results of this meta-analysis.

Isoflavone. Three studies (all BTW-P RCTs) administered isoflavone. The fixed-effect model for these studies was nonsignificant (*g* = 0.120, *z* = 1.401, ns. *Q* (*df* = 2) = 0.905, ns., *I*^2^ = 0.000, tau squared = 0.000). Appendix A summarizes the results of this meta-analysis.

Resveratrol. Two studies (both BTW-P RCT) administered resveratrol, and the results of this fixed-effect model were nonsignificant (*g* = −0.005, *z* =−0.025, ns. *Q* (*df* = 1) = 1.729, ns., *I*^2^ = 42.16, tau squared = 0.085). Appendix A summarizes the results of fixed-effect models tested for RCTs administrating resveratrol.

##### Studies Not Included in Level Three Meta-Analyses

There were four primary studies that were not included in level three meta-analyses because the polyphenol supplementation they had administered did not permit them to be grouped with other studies [20,24,61,62]. Two of these studies had yielded significant effect sizes following the administration of rosmarinic acid [20] and curcumin [62], respectively.

##### Sensitivity Analyses

Each of the level two meta-analyses were repeated, with each study being omitted in turn (i.e., the analysis was repeated with *n* − 1 trial each time) (see Table 3). Three studies in the BTW-P RCT meta-analysis [20,51,62], when removed, led to nonsignificant summary effects, although the result remained marginal in two of these cases. Regarding the crossover studies, none of the omissions of studies led to the generation of a significant summary effect (see Appendix A).

#### 3.6.2. Delayed Retrieval

Sixteen studies were included in the level one random-effects meta-analysis for delayed retrieval. Two studies were excluded from the analysis for the same reasons reported above for the analysis of immediate retrieval [59,63]. The weighted mean effect size for this random-effects analysis was nonsignificant (Hedges’ *g* = 0.022; 95% CI −0.066 (lower) to 0.111 (upper); *z* = 0.499, ns.); *Q* (*df* = 15) = 8.519, ns., *I*^2^ = 0.00, tau squared = 0.00) This result is illustrated by the forest plot in Figure 6. As the result was nonsignificant, only level two meta-analyses were subsequently conducted for delayed retrieval. 

For publication bias, Duval’s and Tweedie’s trim and fill procedure did not trim any studies. Results for Kendal’s tau with continuity correction (tau = −0.141 ns.) and for Egger’s regression procedure (*t* (14) = 0.405, ns.) were evaluated as two-tailed and were nonsignificant. Consequently, these results indicated no evidence of significant publication bias in the results of this level one meta-analysis. A funnel plot for the summary effects in this meta-analysis is shown in Appendix A.

##### Level Two Meta-Analyses for Respective RCT Designs 

Between-participants design RCT (BTW-P). The results of the random-effects model were non-significant (*g* = 0.041, *z* = 0.854, ns.; *Q* (*df* = 11) = 6.95 ns., *I*^2^ = 0.00, tau squared = 0.00). Figure 7 summarizes the results of this meta-analysis.

Crossover trials (CO). The results of the random-effects model were non-significant (*g* = 0.113, *z* = −0.867 ns.; *Q* (*df* = 3) = 0.298, ns., *I*^2^ = 0.000, tau squared = 0.000). Figure 7 also summarizes the results of this analysis. 

##### Sensitivity Analyses

Both level two meta-analyses for delayed retrieval were the subject of sensitivity analyses. None of the omissions of studies led to the generation of a significant summary effect in either case (see Appendix A).

## 4. Discussion

The meta-analyses reported here found a significant effect for polyphenol supplementation for the immediate but not for the delayed retrieval. Furthermore, sub-group analysis by polyphenol groups revealed a positive effect for flavonoids that was only marginally nonsignificant. A distinction was made between these two forms of retrieval to control the effects of polyphenols on any inherent processing differences between them or the neural substrates underpinning them [33,35]. This contribution from cognitive neuropsychological knowledge to the present study is important in facilitating specificity in reporting the results and conclusions. This level of specificity of interpretation is consistent with good practice in meta-analysis [36,37]. By implication, the interpretation of these results is specific to these forms of memory retrieval and should not be generalized to other forms of cognitive tasks. These results were also obtained from primary studies where supplementation was given to participants with a mean age greater than 60 years and who had risk factors for cognitive impairment (i.e., overweight/obese population). These two factors describe the “at-risk” population of interest in the current review. Consequently, the present results should not be generalized to younger populations or to populations with a BMI < 25 kg/m^2^, so that no direct challenge is posed to findings of positive effects of polyphenol supplementation in other populations [18,69]. Our inclusion criteria allowed for the inclusion of studies with participants of any age, so the mean age of participants included suggests a need for research into the possible benefits of polyphenol supplementation for overweight/obese younger people.

The present meta-analytic results support the effectiveness of polyphenol supplementation for immediate retrieval in the at-risk population of interest. Of note is that the level two meta-analyses showed that the studies that had chronic administration (which were all BTW-P) produced a significant summary effect of polyphenols on immediate retrieval compared to studies that had acute administration (all crossover studies). This might imply that chronic administration of polyphenols in the at-risk population could have a beneficial effect on immediate retrieval (including working memory). Additionally, although marginally nonsignificant, chronic administration of flavonoids might be associated with the beneficial effects observed. These findings are consistent with those of another systematic review and meta-analysis by De Vries et al. [70], which reported a significant effect on working and episodic memory, respectively, following chronic administration of polyphenols, specifically flavonoids, anthocyanins, and resveratrol in adults aged 40 and above. In addition, a review by Martain et al. [69] supported a positive effect of cocoa-derived flavonoids (flavanols) on cognitive functions such as memory and executive functions. However, both reviews did not specifically address the at-risk population addressed in the present review (studies reporting overweight/obese participants). Furthermore, De Vries et al. report a publication bias for their results and did not differentiate between working memory and retrieval demands. Thus, it is recommended that future research explores the possible beneficial chronic effects of polyphenols on working memory and other cognitive functions, focusing particularly on flavonoids in at-risk populations (i.e., obese/overweight). 

In the level one meta-analysis for immediate retrieval, three studies yielded a significant summary effect. However, two of these studies [20,62] could not be included for the level three meta-analysis, as no other study had used their form of supplementation. In this context, Rainey-Smith et al. [62] and Herrlinger et al. [20] used curcumin and rosmarinic acid (phenolic acid), respectively, and found a significant effect on immediate retrieval. Recent studies have suggested a possible positive outcome of curcumin supplementation on reducing the risk of Alzheimer’s disease through improving insulin sensitivity and consequently deactivating GSK-3 [71]. This serine-threonine kinase was found to be closely related to hyperphosphorylation of tau and enhanced plaque-associated aggregation of insoluble Aβ when activated [71]. Similarly, anti-inflammatory, antioxidant, and anti-apoptotic activities were linked to the possible neuroprotective effect of rosmarinic acid [72]. Nonetheless, both compounds are under-researched, and their findings in the current meta-analysis point out a potential positive outcome on memory, which needs further investigation in future studies.

Flavonoids were administrated by several studies in this review. However, only chronic administration was found to have a borderline nonsignificant effect on immediate retrieval compared to acute intake. This is consistent with a recent prospective study in older adults who were followed over time, which found an association between a high flavonoid intake and a slower rate of decline in global cognition, episodic memory, semantic memory, and working memory [73]. This indicates the potential for flavonoid intake over an extended period (i.e., years) in preventing cognitive decline. The three studies [21,23,51] yielding a borderline nonsignificant effect administrated flavonoid-rich supplementation in various doses, the smallest being by You et al. [51] (see Table 2). However, none of these three studies reported using a dosage of 500 mg or more per day. This was not consistent with the recommendation by Ammar et al. [25] in a systematic review and meta-analysis on polyphenol-rich interventions with healthy participants, which recommended a daily minimum dose ≥ 500 mg for the possibility of a positive effect of polyphenols on cognition. There is no clear explanation for the trend observed with these three studies given the small dosage they used, but it is notable that they had recruited participants from a population that was at risk of cognitive impairment. Therefore, it is strongly recommended that such a population be included in future flavonoid research. 

The present meta-analyses did not find a significant effect of polyphenol supplementation on delayed retrieval. However, enhanced delayed recall was reported on some measures following acute supplementation with GOE [43] and with chronic resveratrol supplementation [57,60]. As no other study in the sample used GOE, this is a product that may be usefully researched further with this and other at-risk populations. In contrast to Witte et al. [57] and Evans et al. [60], four other studies failed to show any significant effect of chronic resveratrol administration on either immediate or delayed retrieval [55,56,58,59]. The absence of effects on cognition in a younger healthy population has been reviewed by Wightman and Kennedy [18], who suggested that effects of this compound on cognition in more vulnerable populations required investigation. The results of this present review suggest that chronic resveratrol administration made no reliable contribution to delayed memory retrieval performance in the older current at-risk population. 

The evaluation of the primary studies using the Jadad scale [39] showed that details of randomization in the assignment of participants to treatment conditions was the most commonly occurring methodological shortcoming. It is recommended that greater attention be given to the reporting of randomization in future studies. To facilitate future meta-analyses, it is recommended that the means and standard deviations for performance on all cognitive tests administered be made available in tabulated form as a matter of course, either in the main article or as easily accessible supplementary material online.

Although the tight focus of the present review on specific cognitive processes of immediate and delayed retrieval is consistent with good meta-analytic practice regarding the interpretation of results [36,37], it may also be seen as a limitation in the scope of the review regarding the effectiveness of polyphenol supplementation for enhancing cognitive functioning. Different aspects of cognitive functioning draw upon different psychological processes and their supporting neural substrates, so it is recommended that further reviews of findings regarding polyphenol supplementation be conducted on carefully defined areas of cognitive functioning. These may include, for example, semantic memory [74], due to its role in daily functions concerning language. Given the role of planning and decision-making in daily functioning, prospective memory [75] and executive functioning [31] would arguably be other areas of functioning to be explored. 

### Limitations of the Study

There were several limitations for the completion of this study; one of them is the limited RCTs in the field of polyphenols and cognitive functions, specifically memory functions, that included people that were obese/overweight. Furthermore, most of the studies included had an older population, which made it hard to generalize the findings to any other population. Some studies of this systematic review did not report the phenolic composition of their supplementation, which prevented the proper grouping/classification of these studies and consequently their inclusion for the meta-analysis. Furthermore, other studies failed to report statistical information needed for analysis. Finally, the exact distinction between immediate and delayed retrieval memory was not reported by some studies, although the psychological processes underpinning them is different, which led to the exclusion of such studies and consequently the loss of valuable data. 

## 5. Conclusions

The present systematic review and meta-analyses support a potential positive effect of chronic polyphenol supplementation, specifically flavonoids, on immediate retrieval memory in participants aged over 60 years who have obesity, which is a risk factor for cognitive impairment. Such population is under-researched, and further investigation of this potential positive effect is warranted in participants with such risk factors. A potential positive effect for curcumin and phenolic acid was found for immediate retrieval, by which it is recommended to investigate the effects of these compounds further, in addition to flavonoids, in future research. The current study could not establish a conclusion for the recommended dosage for polyphenols to achieve a favorable effect on immediate retrieval. This might be because of the limited research in the field, specifically those that include the obese/overweight population. There was no reliable evidence for a positive effect of polyphenol supplementation on delayed memory retrieval. The current study faced several limitations, such as the failure of some studies to report the phenolic profile of their supplements as well as statistical information that was necessary to be used in the meta-analysis. Furthermore, two important limitations are the limited RCTs that included obese/overweight participants and that these studies had an older population, which made the results of this study limited to that age group. Although consistent with good meta-analytic practice, the study is limited to the cognitive functioning explored (i.e., memory functions), and thus, it is recommended that other areas of cognition, such as executive functioning, be investigated, especially in at-risk populations.

## Figures and Tables

**Figure 1 nutrients-16-00474-f001:**
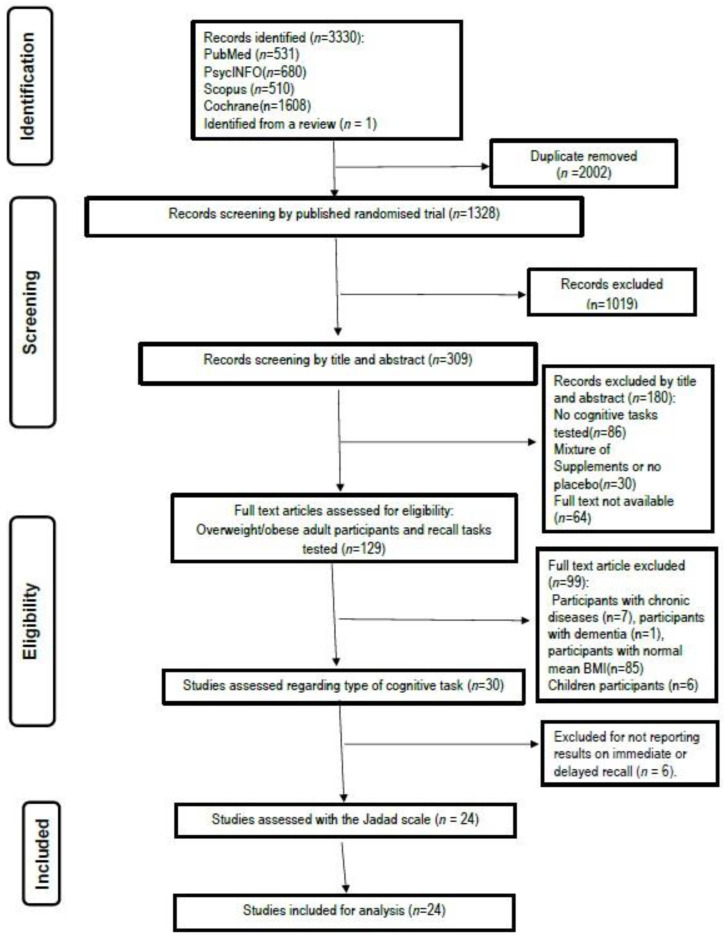
Flow chart of the studies selection process.

**Figure 2 nutrients-16-00474-f002:**
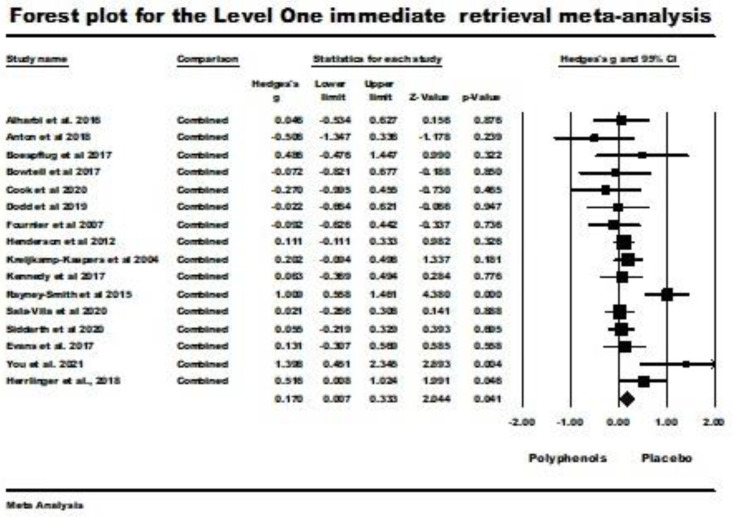
Forest plot and supporting details for the level one immediate retrieval meta-analysis. The comparison column indicates that the mean effect size for all reported comparisons within a study was used to avoid the inappropriate assumption that the comparisons were independent. Data is from the following references [20,21,22,23,24,41,42,43,49,50,51,53,54,58,60,61,62].

**Figure 3 nutrients-16-00474-f003:**
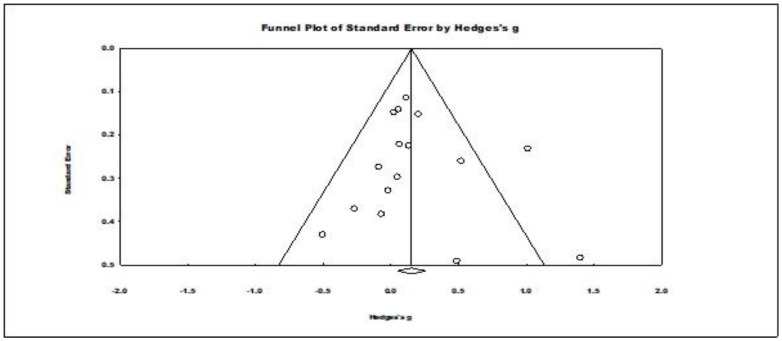
Funnel plot of standard errors by Hedges’ g showing observed effect summary for all the primary studies. The circles in the plot represent studies in the meta-analysis.

**Figure 4 nutrients-16-00474-f004:**
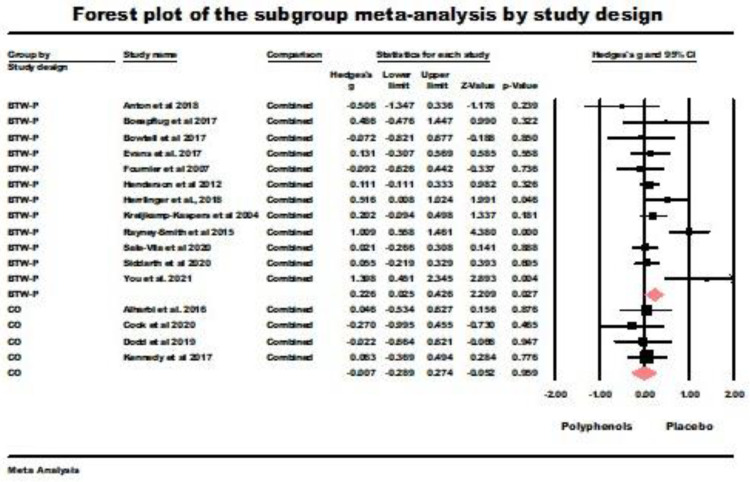
Forest plot of the subgroup meta-analysis for the between-participants RCTs (BTW-P) and crossover studies (CO) using random-effect model for immediate retrieval studies. Data is from the following references [20,21,22,23,24,41,42,43,49,50,51,53,54,58,60,61,62].

**Figure 5 nutrients-16-00474-f005:**
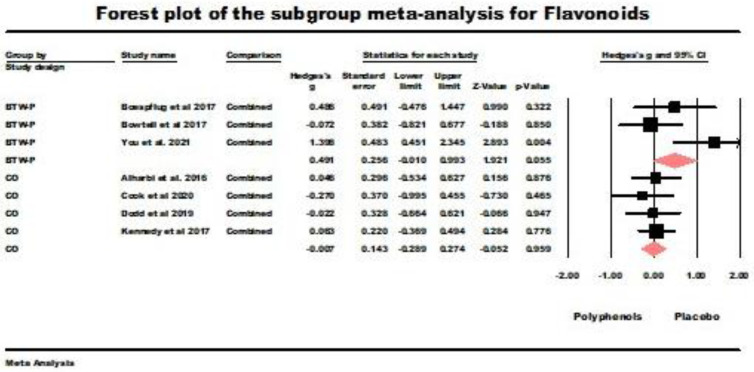
Sub-group meta-analysis for the between-participants (BTW-P) RCTs and crossover studies administering flavonoids. Data is from the following references [21,23,41,43,49,50,51].

**Figure 6 nutrients-16-00474-f006:**
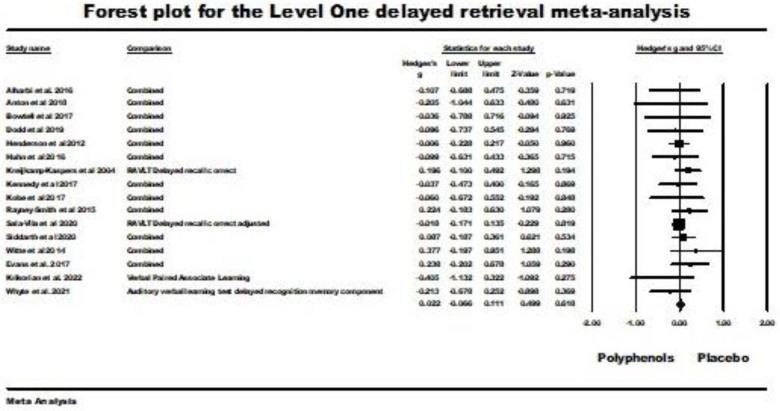
Forest plot and supporting details for the level one delayed-retrieval meta-analysis. Data is from the following references [21,24,41,43,49,52,53,54,55,56,57,58,60,61,62,64].

**Figure 7 nutrients-16-00474-f007:**
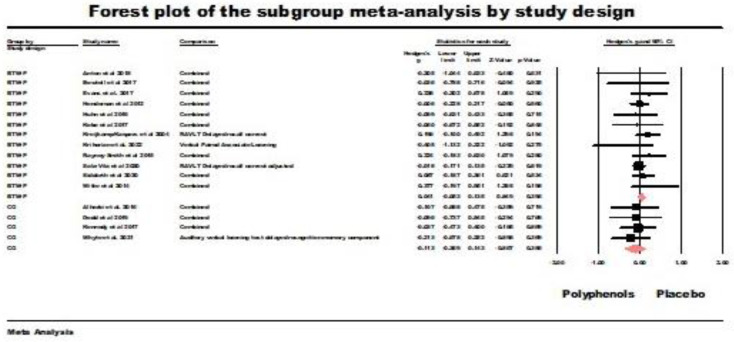
Forest plot of the subgroup meta-analysis for the between-participants RCTs (BTW-P) and crossover studies (CO) using random-effect model for delayed retrieval studies. Data is from the following references [21,24,41,42,43,49,52,53,54,55,56,57,58,60,61,62,64].

**Table 1 nutrients-16-00474-t001:** PICOS (population, intervention, comparison, outcome, and study design) criteria for inclusion of studies.

Parameter	Inclusion Criteria
Population	Obese and/or overweight adults age > 18 (Body Mass Index (BMI) ≥ 25 kg/m^2^)
Intervention	Acute and/or chronic polyphenol-rich supplementation
Comparator	Any: food, juice, placebo
Outcome	Memory function tasks
Study design	Randomized controlled trials

**Table 2 nutrients-16-00474-t002:** Effect of polyphenol-rich supplementation on recall performance.

Author	Design	Intervention	Phenolic Content Information Given in the Articles	Dose, Elapse Time to Cognitive Testing, and Washout	Study Population, Mean (SD if Reported) Age, and BMI	Cognitive Measures	Outcomes for Recall Performance
**Main Polyphenol group: Flavonoids**							
**[21]** Bowtell et al. (2017)	Double-blind, randomized, placebo-controlled trial	Blueberry concentrate	387 mg anthoc yanidins daily. No further information given regarding possible additional phenolic content.	30 mL/day: Chronic administration for 12 weeks before testing: washout N/A	N = 26 Older adults 14 PLA mean age 69.0 years (3.3), BMI 27.1 ± 4.0 kg/m^2^: 12 Blueberry mean age 67.5 years (3.0), mean BMI 25.9 ± 3.3 kg/m^2^	CogState Ltd. cognitive functioning test battery: Groton maze timed chase test and Groton maze learning test (both delayed retrieval); international shopping list task (delayed retrieval): 1-back and 2-back memory tasks (immediate retrieval).	No significant effect found in memory retrieval tasks.
**[23]** Boespflug et al. (2017)	Double-blind, randomized, placebo-controlled trial	Freeze-dried, whole fruit blueberry powder	Daily intake of 417 mg gallic acid equivalents, of which 269 mg were anthocyanins: remainder not specified by type	25 g daily of powder for 16 weeks. Washout N/A	N = 18 Older adults with mild cognitive impairment (MCI) 10 PLA: mean age 75.5 ± 4.8 years, mean BMI 26.4 ± 2.4 kg/m^2^, 8 Blueberry mean age. 80.4 ± 7.3 years, mean BMI 26.2 ± 3.6 kg/m^2^)	N-back task (immediate retrieval)	No significant effect found for memory retrieval.
**[43]** Kennedy et al. (2017)	Double-blind, randomized, placebo-controlled crossover trial	Green oat extract (GOE)	Flavonoid content (calculated as isovitexin) of ≥0.3% (*w*/*w*)	800 or 1600 mg: acute effects on recall tested at baseline and at 1, 2.5, 4, and 6 h post-consumption: 1-week washout between arms of the trial.	N = 42 Healthy participants with subjective memory complaint mean age: 58.90 years (4.8), BMI 25.54 ± 3.18 kg/m^2^	COMPASS word recall (immediate retrieval). Word recall and recognition, names to faces, and picture recognition (delayed retrieval).	A significant positive main effect of treatment on the number of errors made during the delayed word recall task following 800 mg dose GOE.
**[41]** Dodd et al. (2019)	Single-blind randomized, controlled, cross-over trial	Flavonoid-rich blueberry beverage including 300 mL of semi-skimmed milk	579 mg of antho- cyanidins and pro-cyanidins (i.e., combined). No further information given regarding possible additional phenolic content.	30 g of blueberry powder supplement (equivalent to 200 g fresh blueberries): acute effects on recall tested at baseline and at 2 and 5 h post-consumption: washout not reported	N = 18 Adults mean age: 68.7 years (3.3), BMI 25.89 ± 4.46 kg/m^2^	N-back, word recall and recognition, letter memory task (immediate retrieval); word recall and recognition (delayed retrieval).	Only immediate word recognition showed significantly better performance for the intervention condition. No significant effects for any other memory retrieval task.
**[49]** Alharbi et al. (2016)	Double-blind, randomized, placebo-controlled crossover trial	Flavonoid-rich orange drink	272 mg flavonoids	240 mL: acute effects on recall tested at baseline and at 2 and 6 h post-consumption: 2-week washout between arms of the trial.	N = 24 Healthy males mean age 51 years (6.6). Mean BMI 28.3 ± 3.1 kg/m^2^	Immediate word recall (immediate retrieval): delayed word recall (delayed retrieval).	No significant effect found in memory retrieval tasks.
**[50]** Cook et al. (2020)	Double-blind, randomized, placebo-controlled crossover trial	New Zealand blackcurrant extract	105 mg of anthocyanins per 300 mg capsule: daily intake 210 mg. No further information given regarding possible additional phenolic content.	600 mg per day for 7 days. Washout period of 7 days:	N = 14 Older adults mean age 69 ± 4 years, mean BMI 28.5 ± 2.9 kg/m^2^	CANTAB paired associate learning and spatial working memory tasks (immediate retrieval).	No significant effect found in memory retrieval tasks.
**[51]** You et al. (2021)	Double blind, randomized, placebo-controlled trial	Two 250 mg capsules Cosmos caudatus (CC)/day	7.41 Total phenolic content (mg Gallic acid equivalent) (Quercetin 0.9 and Quercitrin 1 (%*w/w*)/100 g)	Chronic administration for 12 weeks before testing: washout: N/A	N = 20 Old adults with MCI 10 PLA Mean age 63.40 ± 2.41 years, mean BMI 25.9 ± 2.7 kg/m^2^ 10 Cosmos caudatus group Mean age 64 ± 4.0 years, mean BMI 26.1 ± 3.19 kg/m^2^	N-back (Immediate retrieval).	No significant effect found in memory retrieval tasks.
**[52]** Whyte et al. (2021)	Double-blind, randomized, placebo-controlled crossover trial	25 g freeze-dried whole wild blueberry (WBB) powder (~1-cup fresh weight)	Total polyphenols (mg) 725 Anthocyanins (mg) 475	25 g powder mixed with water: acute effects on delayed recall tested at baseline, and at 120, 240, 360, and 480 min post-consumption: 1-week washout between arms of the trial.	N = 35 Healthy middle-aged participants Mean age: 50.90 years (7.8), BMI 26.7 ± 4.1 kg/m^2^	Auditory verbal learning test delayed recognition	No significant effect found in memory retrieval tasks.
**Main Polyphenol Group: Flavonoids (Specified as only Isoflavone)**							
**[53]** Henderson et al. (2012)	Double-blind, randomized placebo-controlled parallel trial	Isoflavone-rich soy protein in powder form	91 mg aglycone weight isoflavones	25 g/day: chronic administration for 2.5 years before testing: washout N/A	N = 313 (included in the intention-to-treat analyses) Healthy postmenopausal women 159 PLA: mean age 61 years (7), BMI 26.7 ± 5 kg/m^2^: 154 ISP group mean age 61 years (7), mean BMI 26.5 ± 5 kg/m^2^	California Verbal Learning Test (CVLT) immediate recall, East Boston Memory Test (EBMT) immediate recall, faces immediate recall (all immediate retrieval); CVLT delayed recall, EBMT, and Faces (all delayed retrieval).	No significant effect found in memory retrieval tasks.
**[42]** Fournier et al. (2007)	Double-blind, randomized, placebo-controlled trial	Soya milk Soya supplement	isoflavones 71.6 mg isoflavones 70 mg	353 mL/day 70 mg isoflavones/day For both: Chronic administration for 16 weeks before testing: washout N/A	N = 79 Healthy postmenopausal women 27 PLA, mean age 56.4 (0.8), mean BMI 28.5 ± 1.3 kg/m^2^: 25 Soy milk mean age 56.1 (0.9), BMI 26.8 ± 1.2 kg/m^2^: 27 isoflavone supplement mean age 55.7 (0.7 years), BMI 28.2 ± 0.9 kg/m^2^)	Benton Visual Retention Test (BVRT): visual pattern recognition, forward digit span, corsi block visuospatial recall (all immediate retrieval).	No significant effect found in memory retrieval tasks.
**[54]** Kreijkamp-Kaspers et al. (2004)	Double-blind, randomized, placebo-controlled trial	Isoflavone-rich soy protein in powder form	52 mg genistein, 41 mg daidzein, and 6 mg glycitein: daily total 99 mg. No further information given regarding possible additional phenolic content.	25.6 g/day: chronic administration for 12 months before testing: washout N/A	N = 175 Healthy postmenopausal women included in the modified intent-to-treat analysis. 87 PLA: mean age 66.7 years (4.8), BMI 25.9 ± 3.5 kg/m^2^, 88 Soy group mean age 66.5 years (4.7), BMI 26.41 kg/m^2^	Rey’s auditory verbal learning test (RAVLT) immediate recall, digit span forwards and backwards (immediate retrieval); RAVLT delayed recall (delayed retrieval).	No significant effect found in memory retrieval tasks.
**Main Polyphenol Group: Stilbenes**							
**[55]** Huhn et al. (2018)	Double-blind, randomized, placebo-controlled parallel trial	Resveratrol	200 mg resveratrol and 320 mg quercetin/day:	520 mg daily: chronic administration for 26 weeks before testing: washout N/A	N = 52 Healthy adults 26 PLA mean age 67.54 years (5.07), mean BMI 26.9 ± 4.6 kg/m^2^, 27 Resveratrol group mean age 68.60 years (4.92), mean BMI 26.5 ± 3.8 kg/m^2^	CVLT delayed recall, delayed recognition, and forgetting rate (all delayed retrieval).	No significant effect found in memory retrieval tasks.
**[56]** Köbe et al. (2017)	Double-blind, randomized, placebo-controlled trial	Resveratrol	Resveratrol 200 mg and 350 mg quercetin/day	550 mg daily: chronic administration for 26 weeks before testing: washout N/A	N = 40 Participant with MCI 22 PLA mean age 69 years ± 7, mean BMI 26 ± 3 kg/m^2^ 18 Resveratrol group mean age 65 years ±9, mean BMI 26 ± 3 kg/m^2^	The German version of RAVLT delayed word recall and recognition (all delayed retrieval).	No significant effect found in memory retrieval tasks.
**[57]** Witte et al. (2014)	Double-blind, randomized, placebo-controlled trial	Resveratrol	200 mg of resveratrol and 320 mg quercetin/day	520 mg daily: chronic administration for 26 weeks before testing: washout N/A	N = 46 Healthy overweight BMI 25–30 kg/m^2^ older adults 23 PLA mean age: 63.7 years (5.3) 23 Resveratrol group mean age 64.8 years (6.8)	German version of the AVLT delayed word recall and recognition (delayed retrieval).	Significantly better performance in the supplementation group in the retention, delayed recall, within group significance was found for recognition of words over a 30 min delay.
**[58]** Anton et al. (2018)	Double-blind, Phase IIa randomized, placebo-controlled trial	Resveratrol	Low dose 300 mg per capsule High dose 1000 mg per capsule	Low dose: 300 mg/day of resveratrol. High dose: 1000 mg/day of resveratrol. Chronic administration for 90 days before testing: washout N/A	N = 32 Older adults 10 PLA: mean age: 73.30 years (2.06), mean BMI: 29.74 ± 0.62 kg/m^2^ 12 Low dose: mean age 73.17 years (2.08), mean BMI 29.84 ± 0.62 kg/m^2^: 10 High dose: mean age 73.60 years (2.5), mean BMI 29.03 ± 1 kg/m^2^	Digits forward and backward (immediate retrieval): Hopkins’s verbal learning test-revised (delayed retrieval).	No significant effect found in memory retrieval tasks.
**[59]** Thaung Zaw et al. (2020)	Double-blind, randomized, placebo-controlled trial	Resveratrol	75 mg daily of >98% pure synthetic trans resveratrol	75 mg of resveratrol/day: chronic administration for 12 months before testing: washout N/A	N = 129 Postmenopausal women 66 PLA mean age 64 years (1), mean BMI 25.8 ± 0.5 kg/m^2^ 63 Resveratrol group: mean age 64 years (1), mean BMI 25.4 ± 0.5 kg/m^2^.	NIH-Tool Box: RAVLT immediate word recall (immediate retrieval): RAVLT delayed recall (delayed retrieval).	No significant effect found in memory retrieval tasks.
**[60]** Evans et al. (2017)	Double-blind, randomized, placebo-controlled parallel comparison dietary intervention	Resveratrol	75 mg daily 99% pure synthetic trans-resveratrol (ResVida™)	Chronic administration for 14 weeks before testing: washout: N/A	N = 79 Healthy postmenopausal women 41 PLA mean age 61.5 ± 1.2 years, BMI 26.6 ± 0.8 kg/m^2^ 38 Resveratrol mean age 61.5 ± 1.1 years, BMI 26.8 ± 0.82 kg/m^2^	Visuospatial working memory double span and verbal memory (immediate retrieval), the Rey Auditory Verbal Learning Test (immediate and delayed), the Cambridge Semantic Memory Battery(delayed retrieval).	Significant effect found only for verbal memory and delayed component of the RAVLT.
**Main Polyphenol group: Mixed (Flavonoids and tannins)**							
**[61]** Siddarth et al. (2020)	Double-blind, randomized, placebo-controlled parallel trial	Pomegranate juice	368 mg punicalagins, 93 mg anthocyanins, 29 mg ellagic acid, and 98 mg other tannins: daily total = 588 mg. No further information given regarding possible additional phenolic content.	8 OZ (236.5 mL)/day: chronic administration for 12 months before testing: washout N/A	N = 200 Aging adults with MCI 102 PLA mean age 59.9 years (6.4) years, mean BMI 25.6 ± 4.5 kg/m^2^, 98 Pomegranate group age 60.8 years (6.5), mean BMI 25.3 ± 4.6 kg/m^2^	Brief Visuospatial Memory Test-Revised (BVMT-R): immediate and delayed retrieval) and the Buschke selective reminding test (SRT) for verbal items (immediate and delayed retrieval).	Significant interaction for group X time with no change in learning ability on the BVMT-R (immediate retrieval) for the treatment group versus significant decline for the placebo group over the baseline, 6 months and 12 months testing points. No other significant inter-group effects reported for memory retrieval.
**Main Polyphenol Group: Derivatives of Phenolic acid (a dimer of caffeic acid)**							
**[20]** Herrlinger et al. (2018)	Double-blind, randomized placebo-controlled parallel study	Spearmint extract; 14.5% rosmarnic acid and 24% total polyphenols	600 mg (14.5% rosmarnic acid and 24% total polyphenols) Spearmint extract 900 mg (14.5% rosmarnic acid and 24% total polyphenols) Spearmint extract	600 or 900 mg daily: Chronic administration 90 days before testing: washout: N/A	N = 90 Healthy participants with age-associated memory impairment “30” PLA Mean age 58.2 ± 1.2 years, BMI = 25.9 ± 0.7 kg/m^2^ “30” 600 mg supplement Mean age 59.1 ± 1.0 years, BMI 27.1 ± 0.7 kg/m^2^ “30” 900 mg supplement Mean age 60.8 ± 1.0 years, mean BMI 27.9 ± 0.7 kg/m^2^	Spatial and numeric working memory	Significant improvement in accuracy of spatial working memory and quality of working memory after supplementation with 900 mg spearmint extract.
**Main Polyphenol Group: curcuminoids**							
**[62]** Rainey-Smith et al. (2016)	Double-blind, randomized, placebo-controlled trial	Curcumin (CC)	88% total curcuminoids (curcumin, bisdemethoxycur-cumin, demethoxycurcu-min): 1320 mg per day. No further information given regarding possible additional phenolic content.	1500 mg per day: Chronic administration for 12 months before testing: washout N/a	N = 96 Older adults 57 PLA mean age 65.2 years (6.1), mean BMI 25.8 ± 5.3 kg/m^2^ 39 curcumin mean age 67.2 years (7.2), mean BMI 26.7 ± 3.9 kg/m^2^	RAVLT immediate recall (immediate retrieval): RAVLT delayed recall (delayed retrieval)	No significant effect found in memory retrieval tasks.
**Main Polyphenol Group: Not Applicable**							
**[19]** Lee et al. (2014)	Double-blind, randomized, placebo-controlled trial	Turmeric, cinnamon, or both (CC)	Not separately reported.	Turmeric 1 g, cinnamon 2 g, or both 1 g & 2 g, respectively. Testing at 2, 4, and 6 h from administration: washout N/A.	N = 48 Adults with pre-diabetes Median ages were 71–75 years, Mean BMI 25 kg/m^2^ 12 PLA, 12 Cinnamon 2 g, 12 Turmeric 1 g, 12 Cinnamon 2 g + Turmeric 1 g.	N- back task for digits (immediate retrieval).	Turmeric improved N-back performance over 6 h. Cinnamon was not related to N-back task performance improvement.
**[24]** Sala-Vila et al. (2020)	Parallel-group, observer-blinded, randomized controlled trial	Diet enriched with walnuts	Not separately reported	Walnut diet 30–60 g/day: Chronic administration for 24 months before testing: washout N/A	N = 657 Healthy elders 321 PLA mean age 68.9 (CI: 68.5, 69.3) years, mean BM 27.4 (CI 27.0, 27.9) kg/m^2^: 336 Walnut diet mean age 69.4 (CI 69.0, 69.8) years, mean BMI 27.1 (CI 26.7, 27.6) kg/m^2^	RAVLT immediate word recall and Rey-Osterrieth Complex Figure immediate recall (immediate retrieval): RAVLT delayed word recall (delayed retrieval).	No significant effect found in memory retrieval tasks.
**[63]** Cox et al. (2014)	Double-blind, randomized, placebo-controlled parallel trial	Curcumin (CC) in capsule form	Not separately reported	80 mg per day for 4 weeks. Testing at 1 h and 3 h after administration at the start and finish of the 4 week period: washout N/A.	N = 60 Healthy 30 PLA mean age 69.43 ± 6.579 years mean BMI 27.23 ± 4.818 kg/m^2^: 30 Curcumin mean age 67.56 ± 4.47 years, mean BMI 25.54 ± 3.481 kg/m^2^	COMPASS immediate word recall (immediate retrieval): delayed word recall, delayed word recognition, delayed picture recognition (delayed retrieval).	No significant effect found in memory retrieval tasks.
**[64]** Krikorian et al. (2022)	Double-blind, randomized, placebo-controlled trial	1 packet powder/day freeze-dried blueberry equivalent to 0.5 cc whole-fruit	Not mentioned	Chronic administration for 12 weeks before testing: washout: N/A	N = 27 Healthy adults 13 blueberry group Mean age 55.60 years, mean BMI 31.7 kg/m^2^ 14 placebo group Mean age 57.2 years, mean BMI 33.2 kg/m^2^	Verbal Paired Associate Learning (delayed retrieval)	No significant effect found in memory retrieval task.

Placebo (PLA), Cambridge Neuropsychological Test Automated Battery (CANTAB), Computerized Mental Performance Assessment System (COMPASS), National Institutes of Health Toolbox (NIH-Tool Box), Rey auditory verbal learning test (RAVLT), Brief Visuospatial Memory Test—Revised (BVMT-R), Buschke selective reminding test (SRT), California Verbal Learning Test (CVLT), East Boston Memory Test (EBMT), Benton Visual Retention Test (BVRT, Placebo (PLA), commercial company (CC), green oat extract (GOE), mild cognitive impairment (MCI). N/A = not available.

**Table 3 nutrients-16-00474-t003:** Quality assessment of studies using the Jadad scale (+ means positive score of 1, and – means no score).

Studies	Items of Jadad Scale					
	Randomization Mentioned	Randomization Appropriate	Blinding Mentioned	Blinding Appropriate	An Account of all Participants	Total Score
[49] Alharbi et al. (2016)	+	+	+	+	–	4
[58] Anton et al. (2018)	+	+	+	+	–	4
[23] Boespflug et al. (2017)	+	–	+	+	+	4
[21] Bowtell et al. (2017)	+	–	+	+	–	3
[50] Cook et al. (2020)	+	–	+	–	–	2
[63] Cox et al. (2014)	+	–	+	+	+	4
[41] Dodd et al. (2019)	+	+	–	+	–	3
[60] Evan et al. (2017)	+	+	+	+	+	5
[42] Fournier et al. (2007)	+	–	+	+	+	4
[53] Henderson et al. (2012)	+	+	+	+	+	5
[20] Herrlinger et al. (2018)	+	–	+	–	+	3
[55] Huhn et al. (2018)	+	+	+	+	+	5
[43] Kennedy et al. (2015)	+	+	+	+	+	5
[54] Kreijkamp-Kaspers et al. (2004)	+	+	+	+	+	5
[64] Krikorian et al. (2022)	+	–	+	+	+	4
[56] Köbe et al. (2017)	+	+	+	–	+	4
[19] Lee et al. (2014)	+	+	+	+	–	4
[62] Rainey-Smith et al. (2015)	+	–	+	+	+	4
[24] Sala-Vila et al. (2020)	+	+	+	+	+	5
[61] Siddarth et al. (2020)	+	+	+	+	+	5
[59] Thaung Zaw et al. (2020)	+	+	+	+	+	5
[51] You et al. (2021)	+	+	+	+	–	4
[52] Whyte et al. (2021)	+	–	+	+	+	4
[57] Witte et al. (2014)	+	–	+	+	–	3

## Data Availability

Meta-analysis raw data are available as a supplementary pdf file titled Appendix A: Meta-analysis raw data for immediate recall and Appendix A: Meta-analysis raw data for delayed recall.

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
