# Peer review of "Effect of Polyphenol Supplementation on Memory Functioning in Overweight and Obese Adults: A Systematic Review and Meta-Analysis"

_nutrients, 2024, doi:10.3390/nu16040474_

Round 1

Reviewer 1 Report

Comments and Suggestions for Authors

Review of manuscript reference nutrients-2839497

Title: Effect of Polyphenol Supplementation on memory functioning in Overweight and Obese Adults: A Systematic Review and Meta-Analysis

Authors: Sara Farag, Catherine Tsang, Emad A.S. Al-Dujaili, Philip N Murphy

General comment: the study reviews the potential effect of polyphenol supplementation on cognitive function, especially memory capacity, in overweight and obese adults. Polyphenol supplementation as enhancer of cerebrovascular health, including memory function and overall cognitive capacity, is a topic that has received high attention during the last two decades with incessant information that needs to be organized and summarized. Thus, updated reviews sorting and focusing information are highly necessary and should be welcome. The present review focus interest on one condition such obesity that might be a risk factor for cognitive impairment. The review is very interesting and rightly updated, although it could be better organized. Moreover, a little more focus on flavanols/catechins, which have been recently postulated as significant enhancers of cognitive function, should be expected. Some specific comments are detailed below:

Specific comments:

1)      Line 34; it should be [4,5] instead of [4-5]; the same applies to the rest of the text.

2)      Line 59; it should be [19-21] instead of [19, 20, 21]; the same applies to the rest of the text.

3)      Spelling of several last names in references 9 and 16 should be double checked.

4)      Table 2; table format is not the standard and if data were included within boxes would greatly help understanding of the information. Besides, last names should not be separated in different lines; see reference 23 by Boespflug, reference 21 by Bowtell, ref. 48 by Fournier and others.

5)      Table 2; reference 53, data of dose, etc. It should be mL, capital L. Same for reference 48.

6)      Table 2, the alphabetical order selected for the exposition of studies does not seem appropriate (besides, it is incorrect in references 47 and 60, as well as the last three references); it would be more suitable to group studies associated to specific polyphenol families.

7)      Table 3; alphabetical order also incorrect in reference 47 and last 3.

8)      Since the beneficial effect of flavanols and/or catechins on cognitive function has lately been an attention-grabbing topic, (see reviews by Martín et al. 2020. Effect of Cocoa and Cocoa Products on Cognitive Performance in Young Adults. Nutrients, 12:3691-3704; and Goya et al. 2022. Polyphenols effect on cerebrovascular health. Current Medicinal Chemistry 29:1029-1044), it would be desirable that the authors focused a little more on this particular subfamily of polyphenols widely consumed in tea, win, cocoa and some other foodstuff.

Author Response

Many thanks for the reviewer for appreciation of our work and the valuable feedback provided and here are the responses for the specific comments:

1)      Line 34; it should be [4,5] instead of [4-5]; the same applies to the rest of the text.  

2)      Line 59; it should be [19-21] instead of [19, 20, 21]; the same applies to the rest of the text.

Response: These errors have been corrected as advised.

3)      Spelling of several last names in references 9 and 16 should be double checked.

Response: Corrections have been made to the name as the mistake originated from the referencing software (Refworks).

4)      Table 2; table format is not the standard and if data were included within boxes would greatly help understanding of the information. Besides, last names should not be separated in different lines; see reference 23 by Boespflug, reference 21 by Bowtell, ref. 48 by Fournier and others.

Response: The layout of the table has been made to landscape. The studies in the table were re-organised by polyphenol group (Please see manuscript).

5)      Table 2; reference 53, data of dose, etc. It should be mL, capital L. Same for reference 48.  Always L=Litre is in Capital

Response: corrections have all been made as advised

6)      Table 2, the alphabetical order selected for the exposition of studies does not seem appropriate (besides, it is incorrect in references 47 and 60, as well as the last three references); it would be more suitable to group studies associated to specific polyphenol families.

Response: Table 2 was re-organised (See point 4)

7)      Table 3; alphabetical order also incorrect in reference 47 and last 3.

Response: Table 3 was re-organised, and the alphabetical order was corrected.

8)      Since the beneficial effect of flavanols and/or catechins on cognitive function has lately been an attention-grabbing topic, (see reviews by Martín et al. 2020. Effect of Cocoa and Cocoa Products on Cognitive Performance in Young Adults. Nutrients, 12:3691-3704; and Goya et al. 2022. Polyphenols effect on cerebrovascular health. Current Medicinal Chemistry 29:1029-1044), it would be desirable that the authors focused a little more on this particular subfamily of polyphenols widely consumed in tea, wine, cocoa, and some other foodstuff.

Response: Martín et al. 2020 has highlighted valuable findings regarding the effect of the Cocoa flavonoid: flavanols on cognition in general which support our findings of a potential effect pf chronic flavonoids intake on memory. This has been added in the discussion section to support our findings. However, it should be noted that the population of our study is different from that of Martín et al. 2020 which was also highlighted in the discussion section. As the focus of our systematic review was polyphenols in general it was not possible to focus on subgroups. Nonetheless, as evidence is growing of the effect of flavonoids and its related compounds such as, flavanols and/or catechins on cognition, future original research or systematic reviews might focus on these compounds.

Reviewer 2 Report

Comments and Suggestions for Authors

The manuscript aims to assess the effect of polyphenol supplementation on memory functioning in overweight and obese adults.

Please take into consideration the following suggestions:

Please insert a clear aim of the study at the end of the Introduction section.

Table 2 should be reorganized:

1. Please provide a version of the table which can be easily read and understood (I suggest  you use landscape orientation of the page)

2. Please provide a logical systematization of the table (I suggest you group table entrances by compound)

Please explain the link between polyphenol consumption and the dietary products introduced in table 2 that do not have the polyphenolic content expressed in any way.

I suggest you discuss the relationship between the effects obtained after similar polyphenolic intake.

Please provide a version of table 3 which can be easily read.

Please replace figures 2-7 with figures that can be read.

Please expand the conclusions section.

Please insert a Study limitations section in the manuscript.

Comments on the Quality of English Language

Minor editing of English language required

Author Response

Thank you for your valuable feedback. Please find below the responses for the specific comments:

  • Please insert a clear aim of the study at the end of the Introduction section.

Response: This has been added as advised.

  • Table 2 should be reorganized

Response: The layout of the table has been made to landscape. The studies in the table were re-organised by polyphenol group (Please see manuscript).

  • Please provide a version of the table which can be easily read and understood (I suggest you use landscape orientation of the page)

Response: This has been addressed, Table 2 was re-organised 

  • Please provide a logical systematization of the table (I suggest you group table entrances by compound)

Response: This has been addressed, and the table re-organised by polyphenol group (Please see manuscript).

  • Please explain the link between polyphenol consumption and the dietary products introduced in table 2 that do not have the polyphenolic content expressed in any way

Response: The justification of including studies that did not mention their polyphenol content was added in lines 300-310 in the manuscript.

  • I suggest you discuss the relationship between the effects obtained after similar polyphenolic intake

Response: Studies that used similar supplementation but had reported their phenolic content have been mentioned within the manuscript for instance Rainey-Smith et al reported curcumin composition but Coz et al. did not, both did not report a significant effect. Those that did not report the phenolic content have been outlined as a limitation in the discussion section.

  • Please provide a version of table 3 which can be easily read.

Response: Table 3 has been re-organised, and the alphabetical order was corrected

  • Please replace figures 2-7 with figures that can be read.

Response: All figures have been updated and corrected using PACE

  • Please expand the conclusions section.

Response: This has been addressed as advised. Please see the manuscript.

  • Please insert a Study limitations section in the manuscript

Response: This has been addressed as advised. Please see the limitation section.

Reviewer 3 Report

Comments and Suggestions for Authors

Dear authors,

the topic of your manuscript it quite interesting.

However, I have several remarks:

1. Figure 1 (the PRISMA graphique). Please, provide a better quality figure and a reference.

2. Table 2. Check the punctuation and revise it.

3. Table 3. " Quality assessment of studies using the Jadad scale.": This table is reader-unfriendly. My recomendation is to remove this table and to include this date in the discussion section or to revise the table.

4. Figures 2, 4, 5, 6 and 7. Better quality figures must be provided. This figures are not visible and do not provide any useful information.

5. Figure 3. A better quality figure must be provided or remove the figure and include this information in the main text.

6. Extend the Conclusions.

Author Response

Thank you for your valuable feedback. Please find below the responses for the specific comments:

  • Figure 1 (the PRISMA graph). Please, provide a better quality figure and a reference.

Response: All figures have been updated and corrected using PACE

  • Table 2. Check the punctuation and revise it.

Response: This has been addressed as advised, and the table has been re-organised.

  • Table 3. " Quality assessment of studies using the Jadad scale.": This table is reader unfriendly. My recommendation is to remove this table and to include this date in the discussion section or to revise the table.

Response: Table 3 was re-organised, and the alphabetical order was corrected

  • Figures 2, 4, 5, 6 and 7. Better quality figures must be provided. These figures are not visible and do not provide any useful information.

Response: All figures have been updated and corrected using PACE

  • Figure 3. A better quality figure must be provided or remove the figure and include this information in the main text

Response: All figures have been updated and corrected using PACE

  • Extend the Conclusions.

Response: This has been addressed as advised. Please see the manuscript.
